# Impact of nuclear structure from shell model calculations on nuclear responses to WIMP elastic scattering for $^{19}$F and $^{nat}$Xe targets

**Raghda Abdel Khaleq[1,2⋆], Cedric Simenel[1,2] and Andrew E. Stuchbery[2]**

**1** ARC Centre of Excellence for Dark Matter Particle Physics, Department of Fundamental and Theoretical Physics, Research School of Physics, Australian National University, ACT, 2601, Australia
**2** ARC Centre of Excellence for Dark Matter Particle Physics, Department of Nuclear Physics and Accelerator Applications, Research School of Physics, Australian National University, ACT, 2601, Australia

⋆ raghda.abdelkhaleq@anu.edu.au

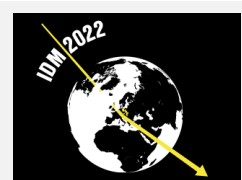

## Abstract

**Non-relativistic effective field theory (NREFT) is one approach used for describing the interaction of WIMPs with ordinary matter. Among other factors, these interactions are expected to be affected by the structure of the atomic nuclei in the target. The sensitivity of the nuclear response components of the WIMP-nucleus scattering amplitude is investigated using shell model calculations for $^{19}$F and $^{nat}$Xe. Resulting integrated nuclear response values are shown to be sensitive to some specifics of the nuclear structure calculations.**

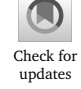

## 1 Introduction

The exact nature of dark matter (DM) continues to elude our understanding, and in response many DM candidates have been proposed [1–10], with Weakly Interacting Massive Particles (WIMPs) being one of the current leading particle candidates. They refer to any new species of particle beyond the standard model which is stable, cold and nonbaryonic. This has encouraged attempts to detect DM-nucleus scattering directly using a variety of nuclear targets [11, 12]. The standard characterisation of the WIMP-nucleus differential cross-section involves both a spin-independent (SI) term and a spin-dependent (SD) one [11, 12]. This

cross-section is often modelled using effective field theories (EFTs) which consider interactions at the energy scale of nucleons, as opposed to the energy scale of standard model quarks. One approach utilises chiral effective field theory (ChEFT), a low-energy effective theory of quantum chromodynamics (QCD) that preserves the QCD symmetries. In particular, one- and two-body WIMP–nucleon currents based on ChEFT have been considered in the case of SI scattering [13–15], as well as in the SD case [16, 17].

Another approach is to form a more complete set of one-body currents using a non-relativistic effective field theory (NREFT) formalism that is independent of the high energy sector. This approach was adopted by [18, 19], and included angular-momentum dependent (LD) as well as spin and angular-momentum dependent (LSD) nuclear interaction responses, in addition to the standard SI and SD ones. Form factors integrated over the momentum transfer $q$ are a proxy for the strength of different interaction channels. These integrated form factor (IFF) values using nuclear shell model wave functions are provided for a range of isotopes relevant for DM direct detection [18, 19]. In the current work, the sensitivity of nuclear IFFs to the nuclear structure of $^{19}$F and $^{nat}$Xe is explored, using the NREFT formalism of [18, 19], with nuclear shell model interactions that differ from those used in the aforementioned works. These nuclear IFF differences are explored in order to quantify the theoretical uncertainties in the overall WIMP-nucleus scattering amplitude due to nuclear structure and modelling.

## 2 Background and methods

### 2.1 DM-nucleus elastic scattering formalism

The relevant NREFT elastic scattering formalism adopted in [18, 19] is briefly outlined here.

The EFT interaction Lagrangian consists of four-field operators of the form [18]

$$\mathcal{L}_{\text{int}} = \sum_{N=n,p} \sum_i c_i^{(N)} \mathcal{O}_i \chi^+ \chi^- N^+ N^-, \tag{1}$$

where $\chi$ represents the dark matter field and $N$ a nucleon field. The non-relativistic operators $\mathcal{O}_i$ in [18, 19] can be used to show that the DM-nucleus elastic scattering amplitude has the form

$$\frac{1}{2J_i+1} \sum_{M_i,M_f} \left| \langle J_i M_f | \sum_{m=1}^A \mathcal{H}_{\text{int}}(\vec{x}_m) | J_i M_i \rangle \right|^2 =$$

$$\frac{4\pi}{2J_i+1} \left[ \sum_{\{j,X\}} \sum_J^\infty |\langle J_i || l_j \, X_J || J_i \rangle|^2 + \sum_{\substack{\{j,X\};\{k,Y\};\{X,Y\} \\ X \neq Y}} \sum_J^\infty \text{Re}\big[ \langle J_i || l_j \, X_J || J_i \rangle \langle J_i || l_k \, Y_J || J_i \rangle^* \big] \right], \tag{2}$$

where $\mathcal{H}_{\text{int}}$ is the interaction Hamiltonian, $J_i$ is the nuclear ground state angular momentum, $A$ is the mass number, and $M_i$ ($M_f$) is the initial (final) angular momentum projection. Here, $X$ and $Y$ are one of six nuclear operators traditionally written as $M_{JM}$, $\Sigma''_{JM}$, $\Sigma'_{JM}$, $\Delta_{JM}$, $\Phi''_{JM}$ and $\tilde{\Phi}'_{JM}$. In the long-wavelength limit ($q \to 0$) $M_{JM}$ is a SI operator, $\Sigma''_{JM}$ and $\Sigma'_{JM}$ are SD, and the remainder are LD, as well as LSD (i.e., spin-orbit and tensor-dependent) operators, respectively. The four DM scattering amplitudes $l_j, l_k \equiv l_{0,E,M,5}$, each associated with a specific nuclear operator $X$, are encoded with the DM and nuclear target physics alongside linear combinations of effective theory couplings [18, 19]. The cross terms in Eq. (2) exist only for two sets of operators, $M_{JM}$, $\Phi''_{JM}$ and $\Sigma'_{JM}$, $\Delta_{JM}$ [18, 19].

## 2.2 Nuclear structure calculations using NuShellX

NuShellX [20] is a nuclear shell model code widely used to calculate nuclear wave functions and common nuclear observables. It is used here to obtain the nuclear inputs for the scattering amplitude in Eq. (2). We employ shell model nuclear interactions which differ from those used in [18,19] and compare both sets of results to test sensitivity to nuclear structure.

## 2.3 Form factors and integrated form factors (IFFs)

A pre-developed Mathematica package [19,21] is used to calculate nuclear form factors

$$F_{X,Y}^{(N,N')}(q^2) \equiv \frac{4\pi}{2J_i + 1} \sum_{J=0}^{2J_i} \langle J_i || X_J^{(N)} || J_i \rangle \langle J_i || Y_J^{(N')} || J_i \rangle \,, \tag{3}$$

where $N, N' = \{p, n\}$. The form factors single out the nuclear aspect of the scattering amplitude. The proton and neutron nuclear operators are given by $X_J^{(p)} = \frac{1+\tau_3}{2} X_J$ and $X_J^{(n)} = \frac{1-\tau_3}{2} X_J$, where $\tau_3$ is the nucleon isospin operator. Only the proton ($N = N' = p$) and neutron ($N = N' = n$) form factor values are evaluated, to isolate differences due to the proton and neutron valence particles. We also consider form factors with $X = Y$ only.

Using a harmonic oscillator single-particle basis, the form factors take on expressions of the form $e^{-y} p(y)$, where $p(y)$ is a polynomial with $y = (qb/2)^2$ and $b$ the harmonic oscillator size parameter. To gauge the numerical strength of these, proton and neutron Integrated Form Factor values are evaluated over a range of $q$, through

$$\int_0^{100 \text{ MeV}} \frac{q\mathrm{d}q}{2} F_{X,Y}^{(N,N')}(q^2), \tag{4}$$

in units of $(\text{MeV})^2$.

## 3 Results

The Integrated Form Factor (IFF) values for alternative shell model interactions are compared in Fig. (1). The $M$ operator IFF values are consistently almost identical between interactions, as the operator coherently takes into account all nucleons inside the nucleus.

### 3.1 $^{19}$F

An unrestricted $sd$ model space is used with single particle levels $1d_{5/2}$, $2s_{1/2}$ and $1d_{3/2}$. The work of [18, 19] uses the USD [22, 23] nuclear interaction, whereas the newer USDB [24] interaction is also employed here. This valence space only allows for positive parity states. These are compared against positive parity experimental levels in Fig. (1).

The largest factor difference in the proton IFF values is 1.03. In the neutron case, the most significant difference between the two interactions is found in the subleading channel $\Phi_n''$ that exhibits a difference of $\approx 20\%$. Although large differences are also found in $\Sigma_n''$ and $\Sigma_n'$, these channels have only small amplitude values, and are thus likely to be irrelevant. The nuclear shell model structure of the valence nucleons can also be used to explain qualitatively the large differences in the proton and neutron values for a given channel, e.g. the $\Sigma''$, $\Sigma'$ channels are proportional to the spin operator $\vec{\sigma}$ in the $q \to 0$ limit, and hence are more sensitive to the

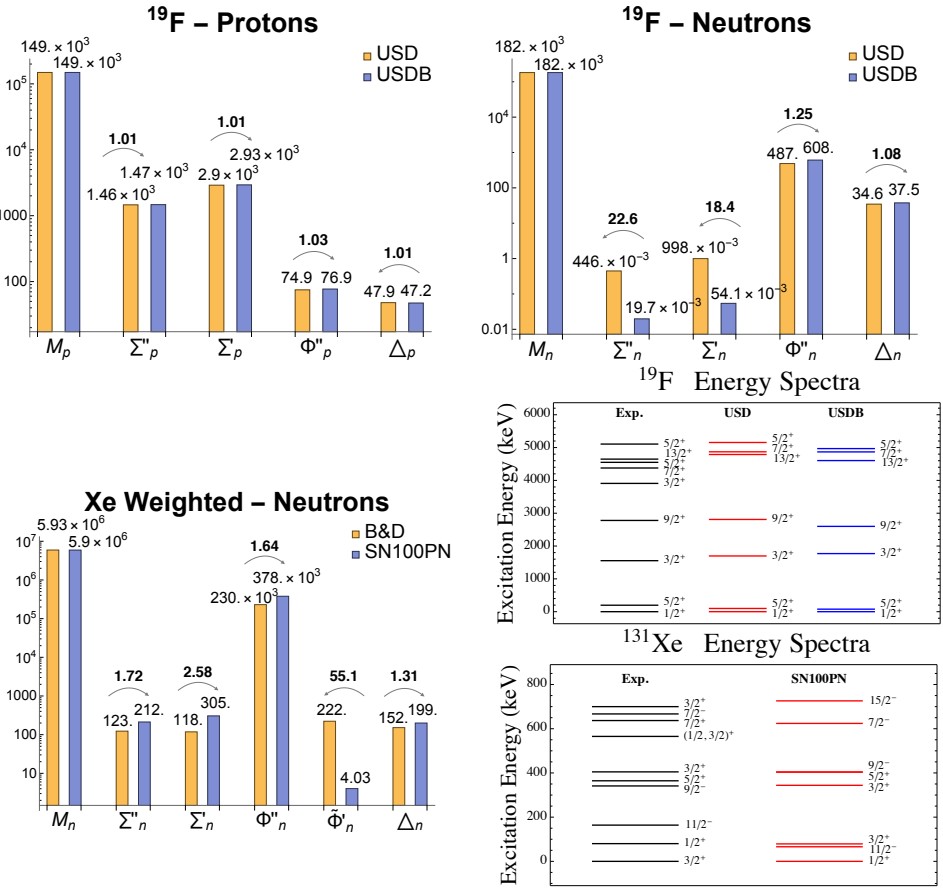

Figure 1: IFF values in units of $(MeV)^2$. Arrows indicate direction of multiplication for the ratio value between the two interactions. Experimental and theoretical energy spectra are also shown (see text).

single unpaired proton in the valence shell, and less sensitive to the two neutrons, which are likely to couple to $J^\pi = 0^+$.

## 3.2 Xe

The model space employed here includes all proton and neutron orbits in the major shell between magic numbers 50 and 82, used alongside the SN100PN interaction [25], whereas the results of [18, 19] use the B&D interaction [26]. We perform unrestricted calculations for $^{131,132,134,136}$Xe, whilst the calculations for $^{129}$Xe and $^{130}$Xe are completed with the proton valence space levels $1g_{7/2}$ and $2d_{5/2}$ unrestricted, and a maximum of 2 protons in the rest. The neutron valence space is unrestricted for the $2d_{3/2}$, $3s_{1/2}$ and $1h_{11/2}$ levels, with a full $2d_{5/2}$ level and a minimum of 6 neutrons in $1g_{7/2}$. To make the $^{128}$Xe calculation more feasible we further restrict the neutron valence space, by keeping the $2d_{3/2}$ and $3s_{1/2}$ levels unrestricted whilst completely filling the $1g_{7/2}$ and $2d_{5/2}$ levels, with a maximum of 8 neutrons in $1h_{11/2}$. A comparison with $^{131}$Xe experimental levels in Fig. (1) shows an inversion of the lowest positive parity states. In this case, the lowest state with the experimental ground state spin parity (e.g., $3/2^+$ for $^{131}$Xe), is used in the calculations.

Most of the proton and neutron interaction channels display non-negligible form factor differences, however only the neutron values are displayed as they are larger in magnitude compared to the proton counterparts. In particular, the subleading operator $\Phi_n''$ shows $\approx 40\%$

difference between the two interactions, which may translate into a large nuclear uncertainty in the overall WIMP-nucleus scattering amplitude given by Eq. (2). The sensitivity of the operator to nuclear structure mirrors the $^{19}$F case, however it is heightened for this heavier isotope.

## 4   Conclusion

To investigate the effect of nuclear structure on the NREFT scattering formalism developed in [18, 19], shell model calculations were performed to obtain the nuclear components of the WIMP-nucleus scattering amplitude for $^{19}$F and $^{nat}$Xe. The shell model nuclear interactions were varied compared to previous work, and the IFF values obtained exhibited comparatively significant differences. These values quantify the strength of each of the nuclear channels considered. To obtain a clearer understanding of the theoretical nuclear uncertainties on the overall WIMP-nucleus elastic scattering amplitude, additional nuclear structure calculations need to be completed, coupled with work linking various high energy models to the NREFT formalism.

## Acknowledgements

**Funding information**   This research was supported by the Australian Government through the Australian Research Council Centre of Excellence for Dark Matter Particle Physics (CDM, CE200100008). RAK acknowledges the support of the Australian National University through the ANU University Research Scholarship.

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
