# Peer review of "Impact of nuclear structure from shell model calculations on nuclear responses to WIMP elastic scattering for $^{19}$F and $^{nat}$Xe targets"

_SciPost Physics Proceedings, doi:SciPost Phys. Proc. 12, 062 (2023)_

## Round 1 · Referee Report · Anonymous · 2022-11-17

Report

This is a rather superficially put together proceedings that describes a nuclear shell model calculation for dark matter scattering form factors on F and natural Xe nuclei. As there is no preceding paper on this calculation authors could point to, it is somewhat important to present matters in a self-contained way. Very little detail is given on the calculation itself and what assumptions are made and the discussion about it should be expanded. In their calculation, does the shell model match on to some known properties of the nuclei involved (such as magnetic moment)?

Also some more words on Eq. (2) would be in order; "l" is not defined.

  • validity: ok
  • significance: ok
  • originality: ok
  • clarity: low
  • formatting: good
  • grammar: good

Author:  Raghda Abdel Khaleq  on 2022-12-23  [id 3181]

(in reply to Report 1 on 2022-11-17)

We thank the referee for their comments on the proceeding. A brief response to the referee's comments is provided below. We hope our proceeding will now be suitable for publication. The authors

1) Details on formalism and calculations

As there is no preceding paper on this calculation authors could point to, it is somewhat important to present matters in a self-contained way. Very little detail is given on the calculation itself and what assumptions are made and the discussion about it should be expanded.

The theoretical modelling of DM-nucleus interaction we employ includes both the NREFT and the nuclear structure modelling with the Shell Model approach. The NREFT formalism is presented in details in Ref [18, 19]. We refer to these works both in the introduction: "In the current work, the sensitivity of nuclear IFFs to the nuclear structure of 19F and nat Xe is explored, using the NREFT formalism of [18, 19]" as well as in the background section: "The relevant NREFT elastic scattering formalism adopted in [18, 19] is briefly outlined here." The limited space of the proceeding does not allow us to present the NREFT formalism in more details here.

All the details required to reproduce the Shell Model calculations are also provided: - The NuShellX code [20] is used. It is a standard Shell Model code that is publicly available and well documented. - The details of the valence space are provided for 19F at the beginning of 3.1 "unrestricted sd model space is used with single particle levels 1d5/2, 2s1/2 and 1d3/2" and for Xe isotopes in the first paragraph of section 3.2: "The model space employed here includes all proton and neutron orbits in the major shell be- tween magic numbers 50 and 82" The details of the valence space truncation are also provided in the same paragraph: "We perform unrestricted calculations for 131,132,134,136Xe, whilst the calculations for 129Xe and 130Xe are completed with ... with a maximum of 8 neutrons in 1h11/2." Finally the details of the different shell model interactions can be found in the references [22,23] (USD), [24] (USDB), [25] (SN100PN) and [26] (B&D), as provided in the manuscript.

This level of detail on the inputs of the Shell Model calculations is standard.

2) Nuclear structure results

In their calculation, does the shell model match on to some known properties of the nuclei involved (such as magnetic moment)?

We agree with the Referee that some details on the nuclear structure results, as predicted by the shell mode calculations, would be beneficial. Although the scattering calculations involve only the ground-state wave-function, the ability to predict the low-lying excited states (energy and spin parity) is an indication that the physical content of the ground state wave function is reasonable. We have therefore added a comparison between the theoretical and experimental spectra in Figure 1. This type of comparison is similar to what is done, e.g., in Ref [14-17]. Due to space limitation, we have added a short description of these plots in paragraph 1 of section 3.1 for 19F:

"The valence space used in 19F calculations only allow for positive parity states. These are compared against positive parity experimental levels in Fig. 1."

and in paragraph 1 of section 3.2 for 131Xe:

"A comparison with 131Xe experimental levels in Fig. 1 shows an inversion of the lowest positive parity states. In this case, the lowest state with the experimental ground state spin parity (e.g., 3/2+ for 131Xe), is used in the calculations."

3) Equation clarification

Also some more words on Eq. (2) would be in order; "l" is not defined.

We have reformulated the last paragraph of 2.1 to clarify the definitions of the four DM scattering amplitudes l, and to avoid confusion with the orbital angular momentum quantum number l.

---

## Editorial Decision

published